# bra-miR167a Targets *ARF8* and Negatively Regulates *Arabidopsis thaliana* Immunity against *Plasmodiophora brassicae*

**DOI:** 10.3390/ijms241411850

**Published:** 2023-07-24

**Authors:** Rujiao Liao, Xiaochun Wei, Yanyan Zhao, Zhengqing Xie, Ujjal Kumar Nath, Shuangjuan Yang, Henan Su, Zhiyong Wang, Lin Li, Baoming Tian, Fang Wei, Yuxiang Yuan, Xiaowei Zhang

**Affiliations:** 1Institute of Horticulture, Henan Academy of Agricultural Sciences, Graduate T&R Base of Zhengzhou University, Zhengzhou 450002, China; 19545630675@163.com (R.L.); jweixiaochun@126.com (X.W.); zhaoyanyan9621@163.com (Y.Z.); sjyang_0614@163.com (S.Y.); 18810835083@163.com (H.S.); nkywzy@163.com (Z.W.); 15037565062@163.com (L.L.); fangwei@zzu.edu.cn (F.W.); 2Henan International Joint Laboratory of Crop Gene Resources and Improvement, School of Agricultural Sciences, Zhengzhou University, Zhengzhou 450001, China; zqxie@zzu.edu.cn (Z.X.); tianbm@zzu.edu.cn (B.T.); 3Department of Genetics and Plant Breeding, Bangladesh Agricultural University, Mymensingh 2202, Bangladesh; ujjalnath@gmail.com

**Keywords:** Chinese cabbage, clubroot, phytohormone, miR167a, auxin

## Abstract

Clubroot is a soil-borne disease caused by *Plasmodiophora brassicae*, which can seriously affect the growth and production of cruciferous crops, especially Chinese cabbage crops, worldwide. At present, few studies have been conducted on the molecular mechanism of this disease’s resistance response. In this experiment, we analyzed the bioinformation of bra-miR167a, constructed a silencing vector (STTM167a) and an overexpression vector (OE-miR167a), and transformed them to *Arabidopsis* to confirm the role of miR167a in the clubroot resistance mechanism of *Arabidopsis*. Afterwards, phenotype analysis and expression level analysis of key genes were conducted on transgenic plants. From the result, we found that the length and number of lateral roots of silence transgenic *Arabidopsis* STTM167a was higher than that of WT and OE-miR167a. In addition, the STTM167a transgenic *Arabidopsis* induced up-regulation of disease resistance-related genes (*PR1*, *PR5*, *MPK3*, and *MPK6*) at 3 days after inoculation. On the other hand, the auxin pathway genes (*TIR1*, *AFB2*, and *AFB3*), which are involved in maintaining the balance of auxin/IAA and auxin response factor (*ARF*), were down-regulated. These results indicate that bra-miR167a is negative to the development of lateral roots and auxins, but positive to the expression of resistance-related genes. This also means that the STTM167a can improve the resistance of clubroot by promoting lateral root development and the level of auxin, and can induce resistance-related genes by regulating its target genes. We found a positive correlation between miR167a and clubroot disease, which is a new clue for the prevention and treatment of clubroot disease.

## 1. Introduction

*Plasmodiophora brassicae* is a soil-borne obligate biotrophic pathogenic protist that infects *Brassica* crops, including Chinese cabbage, cabbage, radish, kale, and broccoli. During infection, *Plasmodiophora brassicae* initially invades from the root hairs and infects the root cortex, causing swollen roots [1,2]. These swollen roots begin to hinder the absorption of water and nutrients, leading to abnormal growth [3,4,5]. In addition, with strong resilience, the pathogenic protist may maintain infectivity for 20 years in the soil, and its dormant spores can exist in the soil for a long time. Infection can start at any time during the cropping season and can affect crop growth and development [6,7,8]. This disease causes severe damage to crop quality and yield, resulting in a 10–15% yield loss of cruciferous plants globally [7,9].

Plant microRNAs (miRNAs) are a group of important endogenous small noncoding RNAs, up to 22 nucleotides in length, which regulate different developmental processes in plants. miRNAs are important regulatory molecules that mediate plant immune responses to biological stress [10]. Several miRNAs have been reported to be involved in plant defense against pathogens [11,12]. When plants are attacked by bacteria and fungi, pathogen-associated molecular pattern-triggered immunity and effector-triggered immunity can protect them against infection [13].

Auxin response factor (*ARF*), a transcription factor, binds to the early responsive elements of auxins, such as the TGTCTC auxin response element, at the promoter, and can mediate auxin signal transduction to regulate plant growth and development [14]. Most *ARF* proteins have three domains: an N-terminal DNA-binding domain; a C-terminal dimerization domain; and a non-conserved intermediate region, known as a transcriptional activation or repression domain. The expression of downstream auxin-responsive factors is regulated by the binding of *ARFs* to *AuxREs* [14,15,16]. *ARFs* play an important role in plant growth and development. To date, 23 *ARFs* have been encoded in the model plant *A. thaliana*. They are involved in the establishment and activity of roots and stem meristems, as well as the formation of embryonic roots, buds, and lateral branches. The function of *ARF* has also been reported in other plants; *ARF6* is involved in the regulation of potato development, with extremely high expression in cells and vascular tissues [17]. *CsARF1* plays an important role in the maintenance and release of deep dormancy for overwintering the buds of tea plants, and *ARF5* plays an important role in the development of secondary xylem in poplar trees [17]. miR167 and its targets, *ARF6* and *ARF8*, also regulate lateral root and adventitious root development. In *Arabidopsis*, miR167 negatively regulates the formation of adventitious roots in the plants, while its target gene *ARF6*/*8* positively regulates the formation of adventitious roots in the plants [18]. Meanwhile, ath-miR167 and *ARF8* are expressed in the pericycle of roots, where lateral roots emerge and mediate the development of lateral roots [18]. In soybean plants, miR167-*GmARF8* is regarded as a key regulatory module of root nodules and lateral development [19].

The miR167 target genes in *A. thaliana* are *ARF6* and *ARF8* [19]. Previous studies have shown that miR167 can regulate the reproductive development processes of *A. thaliana* by regulating the target genes *ARF6/ARF8*, such as stamen and pistil maturation [20]. In addition, miR167 is also involved in the stress response. miR167 targets *BnNRAMP1* in response to cadmium stress in *Brassica napus*. Deletion of the *NRAMP1* gene can increase resistance to cadmium stress in *Brassica napus* [21]. Based on the target mimicry technique, a short tandem target mimic (STTM) was developed [22]. This technique specifically blocks the expression of target miRNAs without affecting other miRNAs [23]. Currently, STTM technology has been widely used in *Arabidopsis*, rice, tomato, soybean, and other miRNA function research [24,25,26,27]. For example, the overexpression of STTM160 and STTM165/166 in *Arabidopsis* [28], rice [29], and cotton [30] inhibit leaf development and increase drought resistance [31]. The introduction of STTM482b into tomatoes can increase disease resistance [32].

Currently, miRNA167 is concentrated in *Arabidopsis*, rice, tomato, and other model plants, and most studies have focused on the regulation of plant growth and development. However, the role of miRNA167 in the development of clubroot disease and regulation of the target genes *ARF6* and *ARF8* has not yet been reported. But previous studies have shown that miR167 is related to plant resistance to biotic stress and can regulate auxin levels in plants [33], which means that miR167 may influence plants’ resistance to clubroot by regulating its target gene *ARF8* to participate in the process of responding to *Plasmodiophora brassicae*. In this study, based on the transcript sequencing result of Chinese cabbage, STTM silencing and overexpression (OE) vectors were constructed and transformed into *A. thaliana*. We analyzed the phenotype and cytology of transgenic *A. thaliana* to explore the function mechanism of how bra-miR167a influences plants’ resistance to clubroot by impacting the plants’ immune systems and auxin levels. In a word, this study could provide a basis on which to further elucidate the role of bra-miR167a in regulating the response to clubroot disease in Chinese cabbage.

## 2. Results

### 2.1. Bioinformatic Analysis of miR167s Reveales the Involvement of Cis-Acting Regulatory Elements

Bioinformatic analysis was performed to preliminarily forecast the biological function of miR167a. There are four precursors of miR167-bra-miR167a/b/c/d. The secondary structures of bra-miR167b and c are the same (Figure 1A). All published sequences of plant miR167 were downloaded from the miRbase database (http://www.miRbase.org/ (accessed on 6 May 2022)) to analyze the conserved bases of miR167. We found that the miR167 family had highly conserved regions at bases 2, 7, 8, and 18 (Figure 1B). A phylogenetic tree was constructed using MEGAX, which showed that pre-miR167a existed both in monocotyledonous and dicotyledonous plants, and bra- miR167a and ath-miR167a were close. This indicates that miR167 is evolutionarily conserved, and that there was high homology between bra- miR167a and ath-miR167a (Figure 1C). To explore the regulatory role of bra-miR167a, we predicted the cis-acting elements in the 2000 bp upstream region of the start codon of miR167a/b/c/d. We found that the bra-miR167 precursor, in addition to containing essential acting elements, also included a jasmonic acid (JA) response element, a plant hormone response element, a stress response element, etc. This means that the expression of bra-miR167a precursors may be induced or inhibited by factors such as light, biotic and abiotic stresses, and hormones (Figure 1D).

### 2.2. bra-miR167a Negatively Regulates the Expression of Targeted ARF and Auxin

To explore the regulation relationship among bra-miR167a, *ARF8*, and auxin levels in plants, we analyzed the targeting sites between bra-miR167a and *ARF8*. The miRNA, transcriptome sequencing, and phytohormone metabolism data were also used to analyze the regulation relationship between bra-miR167a and 31 *ARFs* genes in susceptible materials at four time periods of disease development. We found that target gene *ARF8-b* was directly targeted at 2171 bp and that bra-miR167a was bound to *ARF8-b* in the coding region of the target gene (Appendix A). We found that bra-miR167a was highly expressed at 3 and 9 days after inoculation (DAI), but was low-expressed at 0 and 20 DAI, whereas *ARF8* was highly expressed at 0 and 20 DAI and low-expressed at 3 and 9 DAI (Figure 2A,B). bra-miR167a was shown to negatively regulate its target gene, ARF8. The IAA content was lower in the treated group, which was inoculated with clubroot at 3 and 9 DAI, but the IAA content was further increased at 20 DAI. In the control group, which was not inoculated with pathogen, the IAA content increased in the period of 3–9 DAI, but decreased at 9–20 DAI (Figure 2C). The expression level of bra-miR167a was negatively related to IAA, but positively related to *ARF8* expression, at different time points after inoculation. In conclusion, bra-miR167a negatively regulate *ARF8* gene expression and then auxin content.

### 2.3. STTM167 Negatively and OE-miR167a Positively Regulate the Expression of ARF

To construct the vectors of STTM167a and OE-miR167a, primers were designed for PCR amplification and separated by gel electrophoresis, and the PCR product with a single band and the desired product size was used as a template for synthesizing the sub-clone vector (Appendix A). The amplified STTM167a and OE-miR167a sequences were ligated into the PCAMBIA2301-KY vector at the restriction site KpnI + BamHI (Appendix A). The colony PCR showed that a single band was detected (Appendix A), and the sequencing result showed that the target gene was inserted into the vector (Appendix A).

The transgenic lines were confirmed by PCR amplification using the miRNA167a ST-RT primer [34]. Using quantitative real-time PCR (qRT-PCR), we found that the expression level of bra-miR167a was about 4-fold lower in the STTM167a transgenic plants and 47-fold higher in the OE-miR167a plants than that in the wild type (WT) *A. thaliana*. However, the expression level of *ARF8* increased in the STTM167a transgenic plants, while *ARF8* decreased in the OE-miR167a plants (Figure 3). This result was consistent with the result which we obtained previously, which also proved that bra-miR167a was negatively regulated with *ARF8*. *GUS* staining was also performed to verify the transgenic *A. thaliana*. The staining result of the major roots, lateral roots, and leaves in both STTM167a and OE-miR167a indicated successful transformation (Figure 4).

### 2.4. bra-miR167a Affects Lateral Root Development and Resistance to Clubroot Disease

To determine the role of bra-miR167a in lateral root development, we observed the phenotype in three different periods. From the root phenotype diagram, we can see that WT and STTM had more root development, with a higher number of longer lateral roots. However, OE had obvious main roots and shorter lateral roots, which were fewer in number (Figure 5A,B). To further clarify the specific differences of root development in different transgenic *Arabidopsis thaliana* varieties, we conducted a statistical analysis of the number and length of lateral roots in different transgenic plants. The results showed that the number of lateral roots of STTM167a was 56% higher and the length of lateral roots was 47% higher than that of WT at 15 d after sowing (DAS). OE-miR167a had half the number of lateral roots in transgenic lines that the WT did at 15 DAS, but the length of the lateral roots was not significantly different from the WT type (Figure 5A–F). This result indicates that low expression of bra-miR167a may contribute to the number and length of lateral roots, while its overexpression it may inhibit the number of lateral roots, but is not related to their length.

To investigate the relationship between bra-miR167a and clubroot, we performed phenotypic observation and paraffin section, then calculated the disease index. From the phenotypic picture, there were no obvious differences at 0 DAI between WT, STTM167a, and OE-miR167a, but the roots swelling due to clubroot infection were irregular and varied in size at 30 DAI. The roots of STTM167a did not show significant swelling; OE-miR167a showed significant root swelling at both the lateral and main roots, especially at the junctions of roots and leaves; and WT showed low swelling at both the lateral and main roots (Figure 6A). We also performed paraffin section of root cells at 0 DAI and 30 DAI and found that both WT and OE-miR167a transgenic *Arabidopsis* root cells swelled irregularly at 30 DAI compared to the uninoculated control (Figure 6C). OE-miR167a was filled with a large number of resting spores, whereas STTM167a transgenic *Arabidopsis* showed a normal arrangement of root cells with a small quantity of spores (Figure 6C). Furthermore, the disease index of OE-miR167a *Arabidopsis* was the highest among the WT, STTM167a, and OE-miR167a varieties. The disease index of STTM167a was 58.7% lower than WT and 66.5% lower than OE-miR167a (Figure 6B). These results indicate that transgenic *Arabidopsis* of STTM167a *s* effectively reduces the occurrence of clubroot, while transgenic *Arabidopsis* of OE-miR167a has the opposite function. They also imply that the low expression of bra-miR167a could induce resistance of plants to clubroot.

### 2.5. bra-miR167a Regulate the Expression of Disease-Defense-Related Genes of Transgenic Arabidopsis under the Infection of Clubroot

Plants can not only rely on their own structure and some chemical substances to resist pathogen infection and external environmental stress, but can also use resistance genes (R genes) to monitor and identify pathogen effector factors, causing plants to produce hypersensitive responses (HR) to defend against pathogen invasions. R genes can trigger defense responses, including localized cell death (highly sensitive reactions) and acquired resistance, by encoding R proteins, which are related to the detection of pathogens and pests [35]. Three DAI is a critical period for disease resistance in clubroot [36]; thus, to further understand the mechanism of how bra-miR167a effects clubroots of *Arabidopsis*, four defense-related genes (*PR1*, *PR5*, *MPK3*, and *MPK6*) were selected in order for us to analyze their expression levels in different transgenic materials at 3 DAI and with the control (mock) treatment. We also detected the expression level of bra-miR167a and its target gene, *ARF8*, to explore the regulation relationship between bra-miR67a and the critical resistant genes mentioned above. As the result demonstrated, compared to the control (mock), the expression levels of bra-miR167a and *ARF8* at 3 DAI were significantly lower. The expression level of bra-miR167a of OE-miR167a was significantly higher than STTM167a, while *ARF8* was significantly lower than STTM67a. It indicates that plant could down-regulate miR167a to respond to clubroot stress. Meanwhile, the expression levels of these R genes, *PR1*, *PR5*, *MPK3*, and *MPK6*, in STTM167a transgenic *Arabidopsis* were higher than those in the WT and OE-miR167a transgenic *Arabidopsis* (Figure 7B). The expression levels of *PR1*, *PR5*, and *MPK3* at 3 DAI were higher than those of the uninoculated control. This indicates that the low expression of bra-miR167a reduced the expression of R genes and then activated the plant defense pathway at 3 DAI. Root cross-sections were strained with toluidine blue to determine how the expression level of bra-miR167a impacted *P. brassicae* infection at the lateral stage. STTM167a showed few *P. brassicae* spores, while WT showed obvious spores and OE-miR167a contained numerous enlarged parenchyma cells filled with plasmodia (Figure 7C).

### 2.6. Auxin Signaling Pathway Is Effected in OE-miR167a s under Infection with Clubroot

When plants are stressed by pathogens, auxin can promote the degradation of the indole-3-acetic acid (IAA) transcription repressor family and promote the up-regulation of auxin. The transport inhibitor response 1 (*TIR1*) is an auxin receptor that interacts with IAA protein and can encode the F-box protein that forms IAA [37]. When the concentration of auxin is high, Aux/IAA protein will combine with *SCFTIR1*/*AFB* protein complex and release the *ARF* gene by degrading 26S protease to regulate the transcription of auxin response gene. This then regulates the level of auxin in the plant [38]. Six genes of the *TIR1*/*AFB* protein family were totally expressed, but the expression of each *TIR1*/*AFB* protein was different in different organs. For example, *TIR1*, *AFB1*, and *AFB2* were highly expressed in the root [39]. The target gene, *ARF8*, of bra-miR167a is a key gene in the auxin signaling pathway. To determine how the bra-miR167a regulates the auxin levels of *Arabidopsis* by regulating its target gene *ARF8*, we analyzed the expression levels of the crucial genes *TIR1* (*Bra007720*), *AFB2* (*Bra032954*), and *AFB3* (*Bra026953*) of the auxin signaling pathway to assess the mechanism behind the change in the auxin level (Figure 7C). From the result, we found that the expression levels of *TIR1*, *AFB2*, and *AFB3* in STTM167a transgenic *Arabidopsis* were significantly lower than those in OE-miR167a transgenic *Arabidopsis*, and had significantly decreased compared to the control after 3 DAI. This indicates that low expression of bra-miR167a may up-regulate the expression of auxin protein receptors by up-regulating the expression of *ARF8* to induce resistance in clubroot.

## 3. Discussion

miR167 plays an important role in plant stress response, growth, and development. We found that miR167a plays an important role in lateral root development and root swelling in *Arabidopsis* with clubroot disease. From the bioinformatic analysis, we found that miR167 is conserved during phylogenetic evolution and has the highest homology with *Arabidopsis* miR167a. By data analysis, we found a negative relationship between miR167a and *ARFs*, where miR167a negatively regulated *ARF8-b* and auxin induction (Figure 2).

miR167 also plays a significant role in the response to biotic stresses; for example, miR167 is a positive regulator of nodule and lateral root development in *Glycine max*, which can directly and positively regulate the number of nodules by inhibiting the target genes *GmARF8a* and *GmARF8b*. In rice, Osa-miR167d can negatively regulate resistance to rice blast by regulating its target gene, *ARF12*. In *Arabidopsis*, miR167 is involved in defense against the bacterial pathogen *Pseudomonas syringae*. Overexpression of miR167 results in resistance, inhibition of the auxin response, and stimulation of salicylic acid (SA) signaling [40].

In this study, the silencing vector STTM167a and the OE-miR167a overexpression vector were constructed and transferred into *A. thaliana* using the *Agrobacterium*-mediated method. The expression levels of miR167 and its target genes, *ARF6* and *ARF8*, in different tissues of the transgenic plants were analyzed using qRT-PCR. The expression level of bra-miR167a in STTM167a transgenic plants was significantly lower than that in WT plants, the expression of the target gene *ARF8* was up-regulated, and the number and length of the lateral roots were significantly increased (Figure 5). However, in OE-miR167a plants, the expression of bra-miR167a was up-regulated, the expression of *ARF8* was significantly inhibited, and the number of lateral roots was decreased. This indicated a significant role of bra-miR167a in lateral root development in *A. thaliana*. Similarly, the up-regulation of bra-miR167a in rice caused defects in adventitious roots [41], and the overexpression of bra-miR167a and mutation of *ARF8* lines exhibited a loss of lateral root growth after nitrate stimulation in *Arabidopsis* [42].

bra-miR167a targets *ARF6* and *ARF8* as positive regulators of rooting, whereas miR160 targets *ARF17* as a negative regulator of rooting [43]. In *Arabidopsis*, bra-miR167a can also adopt a root structure by targeting *IAA-Ala Resistance 3* (*IAR3*) under osmotic stress, which can hydrolyzes an inactive form of auxin (indole-3-acetic acid [IAA] -alanine) to release IAA in order to promote root development. Under high osmotic stress, bra-miR167a down-regulation causes the up-regulation of *IAR3*, which promotes lateral root development [44]. This is consistent with our finding that bra-miR167a plays an important role in lateral root development, with an increasing number of lateral roots in STTM167a.

Plants can monitor and identify pathogen effectors by inducing resistance genes and producing a hypersensitive response in order to prevent pathogen invasion [45]. MAPK cascades are key to pathogen invasion in plants. They respond to the coercion process by converting the signals generated from plant cell surface pattern recognition receptors to stress responses [46,47,48]. Among the MAPK cascades, *MPK3* and *MPK6* respond quickly when biological and non-biological coercion are activated [49]. *Pathogenesis-related* (*PR*) genes are important in plants’ responses to stress [50].

Pathogenesis-related (PR) proteins have been recognized as proteins that are strongly induced upon biotic and abiotic stress. Recent progress has revealed that *PR1* can activate the immune response by releasing a C-terminal CAPE1 peptide. Moreover, it can enhance the immune response of the host by forming compounds with *PR5* [51]. It has been reported that *PR1* responds to various stresses, such as drought and *Fusarium* infection [52]. Based on fluorescence quantitative analysis, we found that the expression levels of *PR1*, *PR5*, and *MPK3* in STTM167a transgenic *Arabidopsis* were significantly higher than those in WT and OE-miR167a after 3 d of inoculation (Figure 7B). These results suggest that low expression levels of bra-miR167a could inhibit the development of clubroot disease by inducing resistance genes. Therefore, we suggest that STTM167a transgenic *A. thaliana* is resistant to clubroot. 

In the early stages of pathogen infection, plants transmit and induce disease resistance signals through hormone signal transduction to trigger resistance genes and comprehensively upgrade primary metabolism, energy metabolism, and other metabolic pathways in order to reduce disease severity [53,54]. Transport inhibitor response 1(*TIR1*), an auxin receptor that interacts with AUX/IAA proteins [37], encodes F-box proteins and forms Aux/IAA complexes. F-box proteins form Aux/IAA-SCFTIR1 complexes with SKP1 and Cullin, which are degraded by the ubiquitin/26S proteasome pathway. Aux/IAA proteins can also bind *ARF* to inhibit the transcription of specific auxin response genes [55]. Therefore, we analyzed the expression of the Aux/IAA inhibitor protein receptors *TIR1*, *AFB2*, and *AFB3* and found that the expression of *TIR1*, *AFB2*, and *AFB3* in STTM167a transgenic *Arabidopsis* was significantly lower than that in OE-miR167a transgenic *Arabidopsis* (Figure 7C). The expression levels of *TIR1*, *AFB2*, and *AFB3* were significantly decreased compared with the control at 3 DAI, indicating that STTM167a increased plant resistance to clubroot disease by down-regulating *TIR1*, *AFB2*, and *AFB3*.

Based on these results, a model of the auxin pathway that regulates the pathogenesis of clubroot disease is proposed in this study. When plants are infected by clubroot, the levels of plant hormones like JA, SA and auxin are up-regulated [36]. Then, the changes in plant hormones may up-regulate the auxin protein genes *TIR1/AFB2/AFB3*. In OE-miR167a transgenic *A. thaliana*, the high expression of bra-miR167a negatively regulates the *ARF8*, and then the low expression of *ARF8* promotes the expression of *TIR1*, *AFB2*, and *AFB3*. This inhibits the combination of the auxin/IAA protein with the *ARF8* protein and promotes the degradation of auxin, resulting in the up-regulation of auxin-responsive genes and increased plant susceptibility. In STTM167a transgenic *Arabidopsis*, the low expression of bra-miR167a positively regulates the *ARF8*, and then the high expression of *ARF8* inhibits the expression of *TIR1*, *AFB2*, and *AFB3*, which promotes the combination of the auxin/IAA transcriptional repressor protein with *ARF8* and inhibits the expression of auxin-responsive genes. Meanwhile, in STTM167a transgenic *Arabidopsis*, the resistance-related genes *PR1*, *PR5*, *MPK3*, and *MPK6* were induced at 3 DAI, thus also increasing resistance to clubroot disease (Figure 8). From the schematic diagram, we can obviously determine how bra-miR167a regulates the resistance to clubroot disease. Neither bra-miR167a nor *ARF8* directly regulates auxin levels in plants. bra-miR167a regulates the auxin level by influencing the expression level of *ARF8*, while ARF6/8 regulates it by regulating the response genes of auxin and the change in the expression levels of auxin response genes, which, in turn, effect the transcription levels of the crucial genes. In conclusion, this pattern illustrates the role of bra-miR167a in the response to clubroot disease in detail and provides new insights into the control of clubroot disease.

## 4. Materials and Methods

### 4.1. Sources of Plant Materials and Transformation Vectors

The material BrT24, which is resistant to *P. brassicae*, and the material Y510-9, which is susceptible to it [36], were multiplied in the laboratory of leafy vegetables at the Institute of Horticulture, Henan Academy of Agricultural Science, China. The *P. brassicae* strain used in this study was obtained from a clubroot-infected Chinese cabbage field (*B. rapa*) in Xinye County, Henan Province, China (113.97° E, 35.05° N), and was identified as race 4 by the Williams system [56]. The seeds of BrT24 and Y510-9 were sown into a 50-well tray and placed in an incubator at 25/20 °C with a 16/8 h light/dark cycle and 60% relative humidity. After 20 days of sowing, 20 mL of protist spore fluid of clubroot was inoculated into the roots in each well, and 20 mL sterile water was also injected into the roots of the control group. The root tissue was sampled at 0, 3, and 20 DAI and stored at −80 °C for subsequent experiments.

*Arabidopsis thaliana* accession Columbia (Columbia, 2n) was used for gene transformation. *Arabidopsis thaliana* was first seeded on ½ MS culture plates and, after 10 days, transplanted to a 50-well tray and placed in an incubator at 22/20 °C with a 16/8 h light/dark cycle and 60% relative humidity. After 20 days of sowing, 20 mL of protist spore fluid of clubroot was inoculated into the roots in each well, and 20 mL sterile water was also injected into the roots of the control group. The roots of *Arabidopsis thaliana* were sampled at 0, 3, and 30 DAI and stored at −80 °C for subsequent experiments.

Competent cells of *Escherichia coli* DH5α and *Agrobacterium* GV3101 were purchased from Shanghai Weidi Biological Company (Shanghai, China). The interference and overexpression vectors (pCAMBIA2301-KY (Appendix A)) were provided by Shanghai Kaiyi Biotechnology Co., Ltd. (Shanghai, China).

### 4.2. Prediction of bra-miR167 Target Gene and Bioinformatic Analyses

All plant precursor sequences were downloaded from the miRBase website (http://www.mirbase.org (accessed on 6 June 2022)) and used to construct a phylogenetic tree of miR167 with MEGAX software. The secondary structure was predicted using the RNAfold website, and the conserved regions of miR167 were analyzed using WebLogo (https://weblogo.berkeley.edu/logo.cgi (accessed on 6 June 2022)). The amino acid sequences of miR167 were obtained from Ensemble Plants (https://plants.ensembl.org/index.html (accessed on 6 June 2022)). The conserved domain of miR167 was analyzed using the CDD (Conserved Domain Database CDD) ((http://www.ncbi.nlm.nih.gov/Structuer/cdd/wrpsb.cgi (accessed on 8 June 2022)) of the National Center for Biotechnology Information. The evolutionary tree and protein conserved domains were visualized using TBtools software [57]. The target sites of bra-miR167a and *ARF* were analyzed using psRNATarget (http://plantgrn.noble.org/psRNATarget/ (accessed on 8 June 2022)), with all parameters set as the default. A heat map of bra-miR167a and its ARFs was constructed using TBtools, based on the expression obtained from miRNA and the transcriptome sequence results [58], at 0, 3, 9 and 20 DAI.

### 4.3. Real-Time Quantitative PCR Assay

RNA was extracted using the MiniBEST Plant RNA Extraction Kit (Takara BioTechnology (Dalian, China) Co., Ltd.) from plants, and MonScriptTM miRNA First Strand cDNA Synthesis Kit (Tailing Reaction kit Monad, Suzhou, China) was used for miRNA reverse transcription. MonScript^TM^ RTIII Super Mix, along with the daDNase kit (Monad, Suzhou, China), were used for mRNA reverse transcription. qRT-PCR analyses were performed in Roche LightCycler 480II (Roche, Basel, Switzerland). The program referred to instructions of the qRCR reagent, and the data were quantified using the comparative 2^−∆∆Ct^ method after the PCR program. The expression analysis of each gene was performed with three biological replicates and three technical replicates. Then, the mean of the technical replicates for each biological replicate were calculated, and these values were used for the subsequent statistical analysis. All of the primers which were utilized are listed in Appendix A.

### 4.4. Transformation and Identification of Arabidopsis with STTM167a and OE-miR167a

The STTM vector was constructed by inserting TAG bases into the 10th and 11th bases to silence the mature miRNA sequence. The STTM miR167a intermediate formed a stem–loop structure on its own after being transcribed into RNA, with both ends present as single-stranded complementary portions to miR167a, which binds in a complementary manner to miR167a Synthetic sequences of OE-miR167a and STTM167a target genes were cloned into a pCAMBIA2301-KY (Appendix A) plant expression vector with control of the CaMV35S promoter to generate 35S: bra-miR167a constructs using USER^TM^ cloning technology [59]. These were then transformed, above recombinant constructs, into *Escherichia coli* DH5α. The 35S: bra-miR167a constructs were transformed into Arabidopsis wild-type (Columbia, 2n) plants, and the oral-dip method was used to obtain OEmiR167a and STTM167a transgenic plants [60].

GUS and qRT-PCR were used to identify WT *Arabidopsis* and transgenic lines. The root tissues were sampled at 10 DAS for GUS staining. The tissue was treated with acetone (90%) for 20–30 min at room temperature (25 °C) to remove partial chlorophyll, and then the tissue was fixed with ether for 3 min. Afterward, they were washed with phosphate buffer and soaked in *GUS* staining solution for 8 h at 25–37 °C. Finally, they were transferred into 70% ethanol for decolorization, and this was repeated 2–3 times. The stained samples were observed under a fluorescence microscope and photographed. After 20 days of sowing, qRT-PCR experiments were performed to verify the transform result.

### 4.5. Preparation of Clubroot Spore Solution and Investigation of Disease Incidence

The swollen roots of Chinese cabbage were weighed according to the ratio of target pathogenic fluid volume:swollen root weight = 10:1. They were taken from a −20 °C refrigerator and poured it into a blender to be crushed. The crushed bacterial liquid was filtered, poured into a 50 mL tube, and centrifuged for 5 min (4500 rpm). The supernatant was discarded, and a small amount of sterile water was added into a centrifuge tube and stirred with a glass rod until the precipitate was well mixed. This operation was repeated three times. The supernatant was discarded the final time, and the volume of the supernatant was fixed to the corresponding volume according to the ratio of swelling weight/g:Aseptic water/mL = 1:10. Finally, the concentration of clubroot spore suspension was calculated using a 25 × 16 blood cell counting plate, and the average value was calculated by five-point counting method. The suspension was diluted to 1 × 10^7^ spores/mL to inoculate *Arabidopsis thaliana*.

The roots of the plants were washed, and the incidence of disease was critically observed at 30 DAI. STTM167a, OE-miR167a, and WT were examined, each with three plates and one plate with 50 holes, for a total of 450 plants. Based on the size of the galls, they were graded as follows: grade 0 (no nodule in the lateral and main roots), grade 1 (small nodule in the lateral root), grade 2 (small nodule in the main root), grade 3 (nodule in both the lateral and main roots), and grade 4 (nodule in both the lateral and main roots) [61]. The disease index was calculated as follows: DI = (1 n1 + 2 n2 + 3 n3 + 4 n4) × 100/4 Nt (n represents the number of diseased plants/grade, and N represents the total number of diseased plants): DI = 0 (highly resistant), DI < 10 (resistant), 10 ≤ DI < 20 (moderately susceptible), 20 ≤ DI < 50 (susceptible), and DI ≥ 50 (highly susceptible). For each plate, a disease index was calculated, and each line of *Arabidopsis thaliana* received three disease indexes.

### 4.6. Paraffin Sectioning

Fresh tissue samples measuring about 1 cm was cut and placed it into a fixative solution. Then, they were taken from the solution, and the tissue was flatted and put into the embedding frame. Thereafter, the samples were placed in a dehydrating box into a dehydrator, wax dipping was performed according a series of treatments (75%—4 h; 85%—2 h; 90%—2 h; 95%—1 h; 100%—1 h; alcohol benzene—10 min; xylene—20 min; −65 °C—3 h). First, the melted wax was placed into the embedding frame before the tissue was removed and also placed into the embedding frame. After cooling on a −20 °C freezing platform, the solidified wax was removed from the embedding frame for flatting. The flattened wax block was sliced with a thickness of 4 μM. Slices were placed on a spreading machine to flatten the tissue, then placed in a 60 °C oven for baking. The baked wax was removed, and the sample was stored at room temperature for later use. Slices were placed into toluidine blue dye solution for about 2–5 min, then washed with water. Finally, microscopic inspection was conducted.

### 4.7. Data Analysis

The miRNA, transcriptome sequencing, and phytohormone metabolism data analysis methods referred to Xiaochun Wei [36]. All three repeated biological experiments were analyzed using IBM SPSS Statistics 2.6, and one-way or two-way variance analysis (ANOVA) was performed to determine the significance of the statistics. The given value is the average ± standard deviation (SD) for three biological replicates, and the SD values are shown by the error bars in the figures. The *p* value is corrected using Bonferroni and indicated by * *p* < 0.05, ** *p* < 0.01, *** *p* < 0.001, **** *p* < 0.0001. Moreover, different letters indicate significant differences at *p* < 0.05.

## 5. Conclusions

In this study, we explored the molecular mechanism by which bra-miR167a influences resistance to clubroot by bioinformatically analyzing miR167 and investigating the OE-miR167a and STTM167a. From the study’s results, we found that the number of lateral roots of STTM167a was significantly higher than that of OE-miR167a and WT, and the disease index of STTM167a was also significantly lower than that of OE-miR167a and WT, indicating that bra-miR167a was negatively correlated with both the number of lateral roots and resistance to clubroot disease. To further explore this, we detected the expression level of the resistance genes and auxin receptor genes. The results showed that the expression levels of the resistance genes of STTM167a were higher than those of OE-miR167a and WT, while the auxin receptor genes was even lower. This means that the low expression of bra-miR167a is able to up-regulate the resistance genes *PR1*, *PR5*, *MPK3*, and *MPK6* and down-regulate the auxin/IAA auxin-inhibitory protein receptor genes *TIR1*, *AFB2*, and *AFB3* to improve resistance to clubroot disease. In conclusion, bra-miR167a regulates the auxin pathway and responds to the immune system by regulating the target gene *ARF8*, the homeostasis of plant hormones, and resistance to clubroot disease. This indicates that we may be able to improve plant varieties by silencing or overexpressing the crucial miRNAs of some biological and abiotic stresses. However, our study simply clarified the mechanism of bra-miR167a-*ARF8* in *P. brassicae* infection, but the deep interaction mechanism among miR167a, the immune system, plant hormones, and more miRNAs related to other biological and abiotic stresses should be studied further.

## Figures and Tables

**Figure 1 ijms-24-11850-f001:**
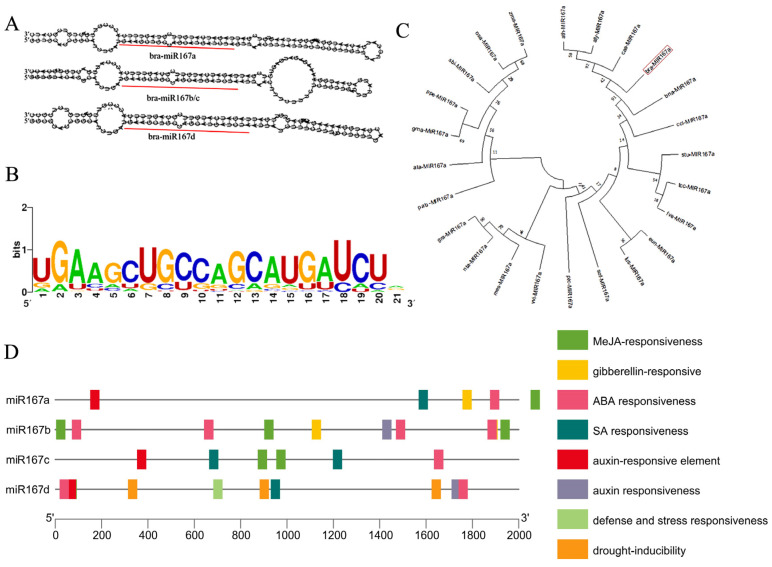
Bioinformatic analysis of miR167s. (**A**) Predicted secondary structure of miR167 precursor. (**B**) Base conservation analysis of miR167 mature sequence. (**C**) Phylogenetic tree based on miR167a precursor sequence. (**D**) Cis-elements analysis of cis-regulatory elements in bra-miR167.

**Figure 2 ijms-24-11850-f002:**
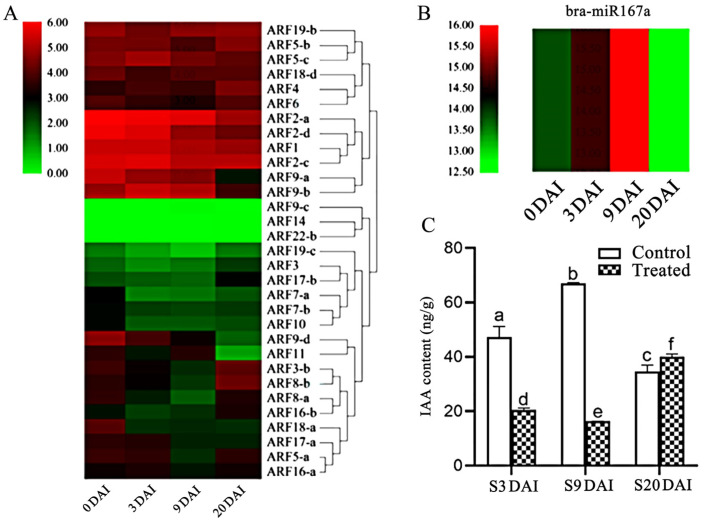
Analysis of the expression patterns of bra-miR167a and *ARFs* in Chinese cabbage. (**A**) The heatmap analysis of *ARFs* expression; (**B**) the heatmap analysis of miR167a in susceptible materials at different periods after inoculation. (**C**) IAA content was measured in the treated group, which was inoculated with clubroot pathogen, and the control group was not inoculated for susceptible materials at different periods. The different lowercase letters indicate significant differences (*p* < 0.05) based on Duncan’s test.

**Figure 3 ijms-24-11850-f003:**
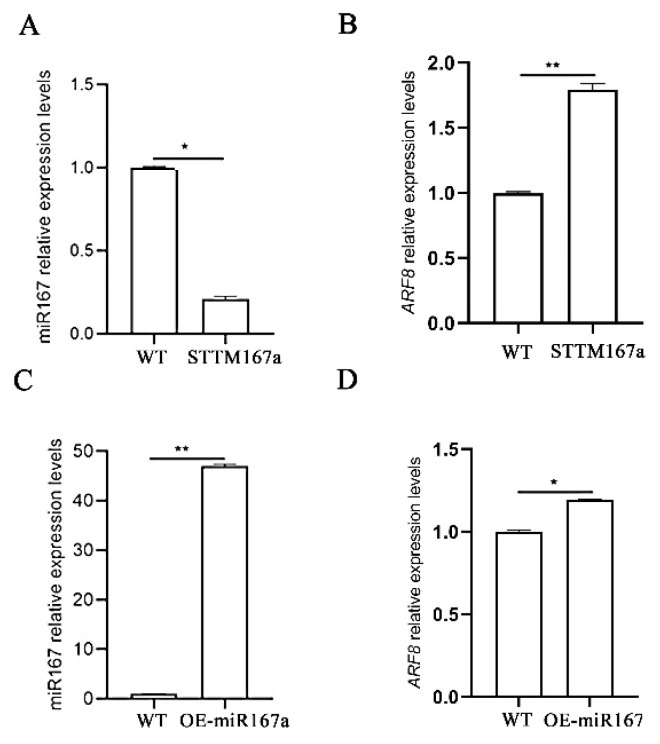
Expression analysis of miR167a and *ARF8* in STTM167a and OE-miR167a. (**A**) Analysis of miR167a and (**B**) *ARF8* expression analysis in STTM167a plants; (**C**) analysis of miR167a and (**D**) *ARF/8* expression analysis in OE-miR167a plants. Asterisks (* *p* < 0.05, ** *p* < 0.01) indicate significance based on *t* test (*p* < 0.05).

**Figure 4 ijms-24-11850-f004:**
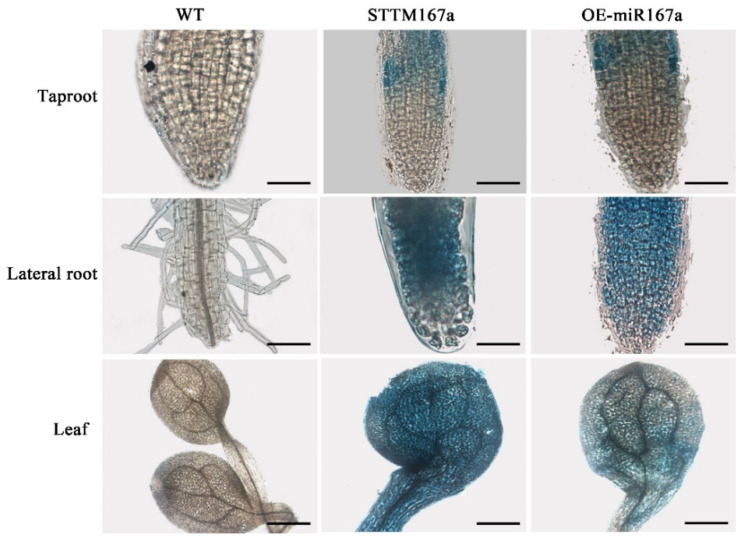
GUS staining analysis of different tissues of WT, STTM167a, and OE-miR167a transgenic *Arabidopsis*. bar = 1 mm.

**Figure 5 ijms-24-11850-f005:**
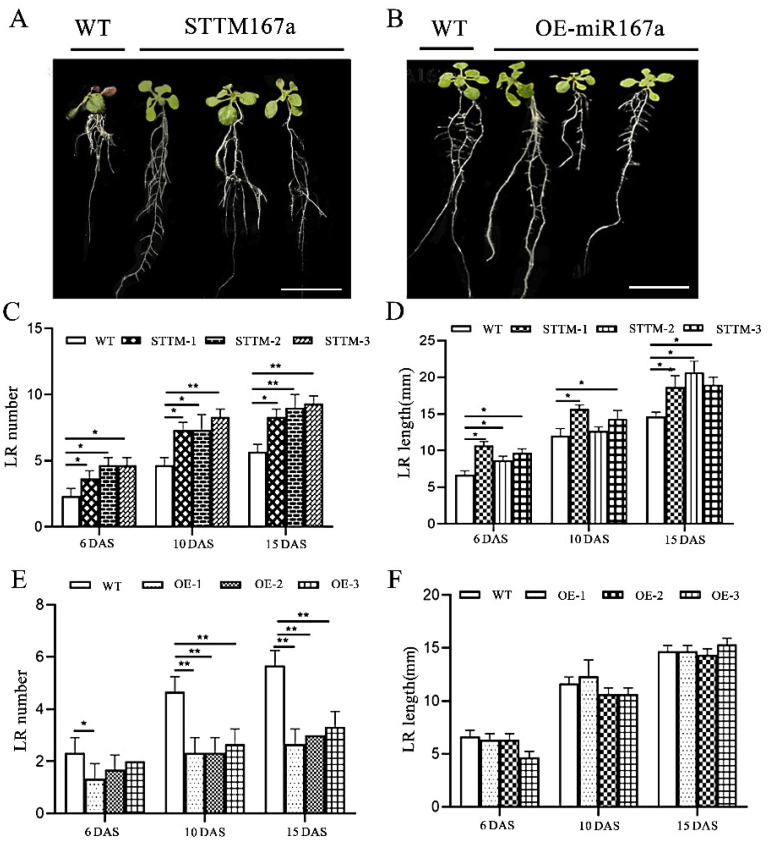
Lateral root analysis of STTM167a and OE-miR167a transgenic plants. (**A**): STTM167a and (**B**): OE-miR167a root phenotype analysis at 20 DAS (bar = 50 mm). (**C**): STTM167a lateral root number analysis. (**D**) STTM167a lateral root length analysis. (**E**) OE-miR167a lateral root number analysis. (**F**) OE-miR167a lateral root length analysis. Asterisks (* *p* < 0.05, ** *p* < 0.01) indicate significance based on *t* test (*p* < 0.05).

**Figure 6 ijms-24-11850-f006:**
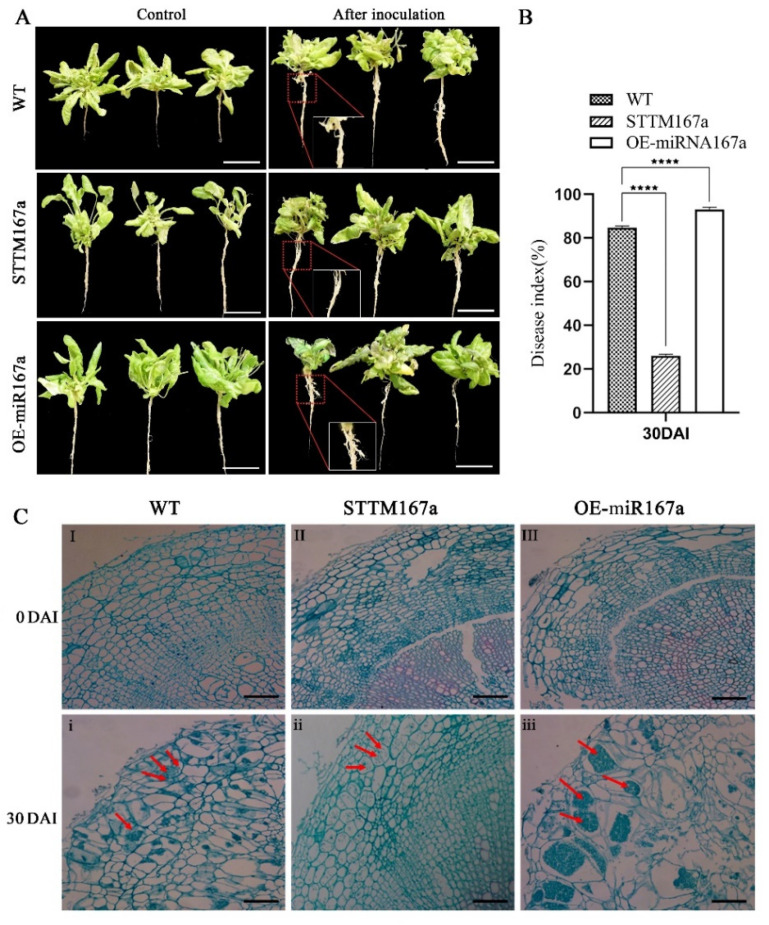
The disease status of WT, STTM167a, and OE-miR167a *Arabidopsis thaliana* plants at 30 days after inoculation with *P. brassicae* (**A**) Root phenotype analysis of WT, STTM167a, and OE-miR167a *Arabidopsis thaliana* plants at 30 DAI with *P. brassicae.* (**B**) Disease index statistics (bar = 50 μm). (**C**) Observation of paraffin sections of WT, STTM167a, and OE-miR167a roots at 30 DAI. **I** and **i** are the cross-sections of the WT control, both uninoculated and at 30 DAI, respectively; **II** and **ii** are STTM167a, both uninoculated and at 30 DAI; **III** and **iii** are cross-sections of paraffin sections of the OE-miR167a control, both uninoculated and at 30 DAI, respectively (bar = 50 μm). The red arrow indicates the *P. brassicae* spores. Asterisks (**** *p* < 0.0001) indicate significance based on t test (*p* < 0.05).

**Figure 7 ijms-24-11850-f007:**
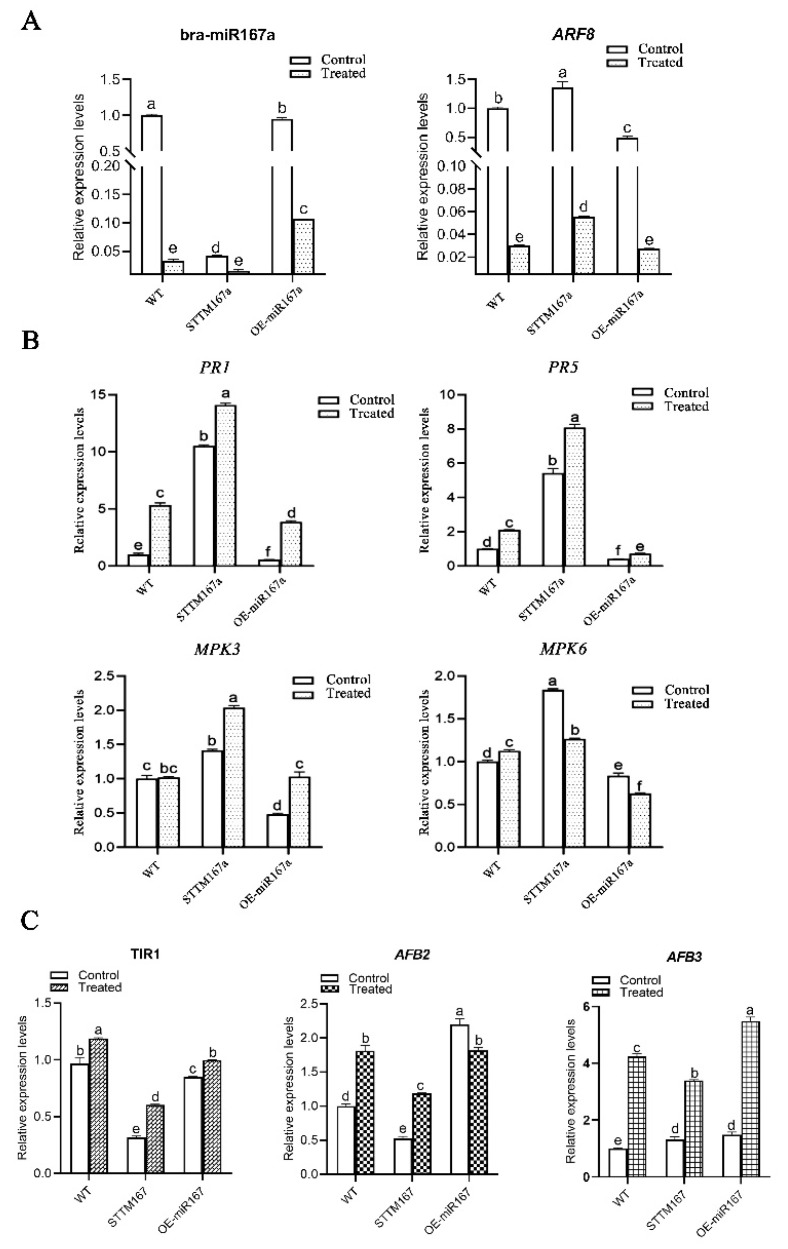
(**A**) Expression changes of bra-miR167a and its target gene, *ARF8*, in different transgenic plants; (**B**) expression changes of disease-resistance-related genes in different transgenic plants; and (**C**) expression changes of *TIR1*, *AFB2*, and *AFB3* in different transgenic plants at 3 DAI. The expression value at 3 DAI of WT of control was normalized as 1. The different lowercase letters indicate significant differences (*p* < 0.05) based on Duncan’s test.

**Figure 8 ijms-24-11850-f008:**
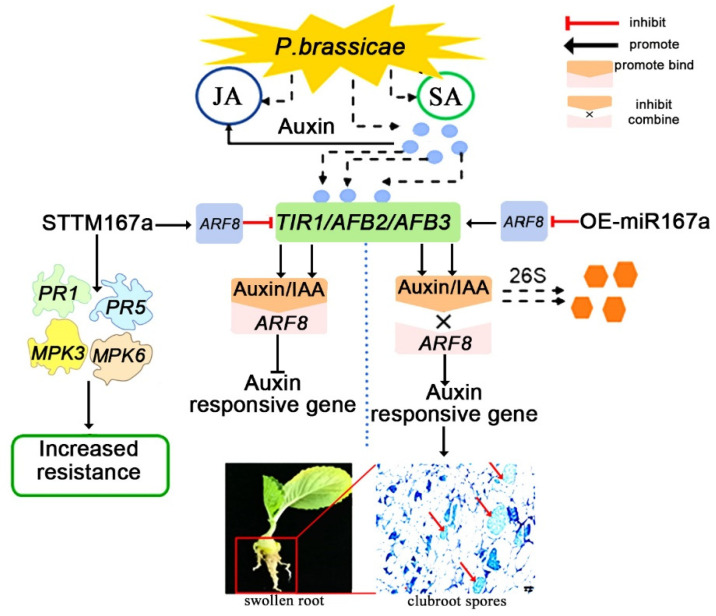
A schematic diagram of the involvement of auxin in the process of clubroot disease. Auxin response factor 8 (*ARF8*), pathogenesis-related gene 1 (*PR1*), pathogenesis-related gene 5 (*PR5*), map kinase 3 *(MPK3*), and map kinase 6 (*MPK6*).

## Data Availability

The original contributions presented in this study are publicly available. The data can be found in the National Center for Biotechnology Information (NCBI) BioProject database under accession numbers PRJNA868821 and PRJNA743585.

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
