# Peer review of "bra-miR167a Targets ARF8 and Negatively Regulates Arabidopsis thaliana Immunity against Plasmodiophora brassicae"

_ijms, 2023, doi:10.3390/ijms241411850_

Round 1
Reviewer 1 Report
The defeat of plants of the cruciferous family with clubroot leads to a violation of the growth and development of plants and causes serious harm to agriculture. The authors of the submitted manuscript offer one of the options for protecting a plant (using the example of Arabidopsis) from being damaged by clubroot. The results and conclusions obtained are beyond doubt. The manuscript is carefully formatted, but there are a few notes on the design of references: several references have no pages indicated, reference 51 does not have a journal.
316-R genes. What are these genes?
Figure 5 A and B - what is the difference between transgenic plants. Specify in caption
Figure 6 C - indicate contrasting designations in the figure. Indicate the designation of red arrows in the caption
Author Response
Dear Reviewer,
I would like to extend my gratitude to the respected reviewers, who spend your valuable time to review our manuscript. Your valuable comments were necessarily important for substantial improvement of our manuscript. The given comments and suggestions which we try to address point-by- point as follows:
Point 1: The defeat of plants of the cruciferous family with clubroot leads to a violation of the growth and development of plants and causes serious harm to agriculture. The authors of the submitted manuscript offer one of the options for protecting a plant (using the example of Arabidopsis) from being damaged by clubroot. The results and conclusions obtained are beyond doubt. The manuscript is carefully formatted, but there are a few notes on the design of references: several references have no pages indicated, reference 51 does not have a journal.
316-R genes. What are these genes?
Response 1: Thanks for your comments. We have corrected this error. See in line 630 “Zhang, S.Klessig, D.F. MAPK cascades in plant defense signaling. Trends in Plant Science 2001, 6 (11), 0-527.”
Point 2: 316-R genes. What are these genes?
Response 2: Thanks for your comments. We have added this content in line 209-214:” plants can not only rely on their own structure and some chemical substances to resist pathogen infection and external environmental stress, but also use resistance genes (R genes) to monitor and identify pathogen effector factors, causing plants to produce hypersensitive responses (HR) to defend against pathogen invasion. R genes can trigger defense responses including localized cell death (highly sensitive reactions) and acquired resistance by encoding R proteins which are related to the detection of pathogens and pests”.
Point 3: Figure 5 A and B - what is the difference between transgenic plants. Specify in caption
Response 3: Thanks for your comments. We have added this content in line 171-176:” From the root phenotype diagram, we can see WT, STTM had more root development with longer lateral root in length with higher numbers. However, OE had obvious main roots, shorter lateral roots with fewer numbers (Figure 5A and B). To further clarify the specific differences of root development in different transgenic Arabidopsis thaliana, we conducted statistical analysis on the number and length of lateral roots in different transgenic plants.”
Point 4: Figure 6 C - indicate contrasting designations in the figure. Indicate the designation of red arrows in the caption
Response 4: Thanks for your comments. We have modified this in line 230-234 “Root cross section were strain with toluidine blue to figure out how the expression level of bra-miR167a impacts P. brassicae infection at lateral stage. STTM167a showed few P. brassicae spores while WT showed obvious spores and OE-miR167a contained numerous enlarged parenchyma cells filled with plasmodia (Figure 7C).” and 235-242:” IAA content determination of treated group was inoculated with clubroot pathogen and control group was not inoculated by pathogen for susceptible materials at different periods.”

Reviewer 2 Report
Liao and collaborators aim to investigate the role of mir167 in immunity against P. brassicae. They started from information and transcriptomic data from Chinese cabbage that indicate the possible involvement of the mir167-ARF8 module in response to clubroot disease. The authors performed bioinformatic and comparative analysis on the conservation of bra-miR167, they tried to validate the hypothesis in Arabidopsis and suggest possible molecular pathways influenced by ARF8 to promote resistance to P. brassicae.
The research question is relevant and meaningful and it has the potential to bring would new information for research on Brassica crops, critical commercial crops. The introduction and discussion section is well written and it is interesting the model proposed, but, unfortunately, the data are still too preliminary to support conclusions. In particular, I have several concerns about the rationale of the experiments conducted in Arabidopsis. Here are my main concerns.
Main concerns:
- It is not clear the origin of transcriptomic data of Chinese cabbage (the mentioned ref. 33 Liu et al., 2021 seems a review paper), and how the analysis was performed. On which samples and tissues? Integrating ARF8 in-situ hybridization on Chinese cabbage root tissue would be nice.
- I was wondering why the authors did not choose to check the involvement of Arabidopsis endogenous mir167a and ARF8 in immunity by directly performing analysis using the vast amount of tools already available (single arf8 mutant, double arf6/8, 35S:mir167). On the contrary, they chose to overexpress bra-miR167 (the sequence homology between ath-mir167 and bra-miR167 is not reported nor is the conservation of bra-miR167 on the binding on Ath-ARF8). If the idea was any way to obtain the silencing of ARF8 why did they not choose to overexpress ath-mir167 directly?
- Related to this, It is not clear whether the STTM167a lines carry the binding domain of bra-miR167a or ath-miR167a.
- The possible effect on ARF6 expression is never mentioned nether tested. Might the overexpression of bra-miR167 in Arabidopsis repress unspecific targets?
- Authors are also missing to report in the introduction about the information already available in the literature regarding the role of ARF8 and ARF6 in adventitious root development. The root phenotype reported in Figure 5 should be implemented with a detailed analysis regarding the effects on the adventitious roots (ref 44 is mentioned only after in the discussion section).
- All section 2.2 of the results are mainly material and method information.
- Section 2.4 Figure 6. This part should be the most important, but the experiment conducted to test susceptibility to clubroot is poorly explained (how many biological replicates were done on how many plants?) Is the result reported in Fig. 6B statistically significant? Which statistical test was conducted?
moderate/minor editing
Author Response
Dear Reviewer,
I would like to extend my gratitude to the respected reviewers, who spend your valuable time to review our manuscript. Your valuable comments were necessarily important for substantial improvement of our manuscript. The given comments and suggestions which we try to address point-by- point as follows:
Point 1: It is not clear the origin of transcriptomic data of Chinese cabbage (the mentioned ref. 33 Liu et al., 2021 seems a review paper), and how the analysis was performed. On which samples and tissues? Integrating ARF8 in-situ hybridization on Chinese cabbage root tissue would be nice.
Response 1: Thanks for your comments. We have corrected the references and provided a detailed explanation of the cabbage materials in the materials and methods in line 374-380 ” Resistant material BrT24 and susceptible material Y510-9 to P. brassicace [37] were multiplied in the laboratory of leafy vegetables at the Institute of Horticulture, Henan Academy of Agricultural Science, China. P. brassicae strain used in this study was obtained from a clubroot-infected Chinese-cabbage field B. rapa in Xinye County, Henan Province, China (113.97°E, 35.05°N) and was identified as race 4 by the Williams system [59]. The seeds of BrT24 and Y510-9 were sown into a 50-well tray and placed an incubator at 25/20 °C with 16/8 h (light/dark) and 60% relative humidity. After 20 days of sowing, 20 ml of protist spore fluid of clubroot was inoculated into root in each well and 20 ml sterile water was also injected into in the control group. The root tissue was sampled at 0, 3, 20 DAI and stored at -80 °C for subsequent experiments”
Point 2: I was wondering why the authors did not choose to check the involvement of Arabidopsis endogenous mir167a and ARF8 in immunity by directly performing analysis using the vast amount of tools already available (single arf8 mutant, double arf6/8, 35S: mir167). On the contrary, they chose to overexpress bra-miR167 (the sequence homology between ath-mir167 and bra-miR167 is not reported nor is the conservation of bra-miR167 on the binding on Ath-ARF8). If the idea was any way to obtain the silencing of ARF8 why did they not choose to overexpress ath-mir167 directly?
Response 2: Thank you so much for asking such a professional question. We chose bra-miR167 as the main study material partly because our previous results indicated that bra-miR167 was involved in the process of resistance and susceptibility of clubroot disease. The reason why bra-miR167a was overexpressed and silenced was mainly because the aim of this experiment was to explore the mechanism of bra-miR167a's resistance to clubroot disease, how does bra-miR167a affect the resistance of plants to clubroot when the expression level of bra-miR167a changes. bra-miR167a was chosen over ath-miR167a because bra-miR167a had been proved to have an effect on clubroot disease, because miR167 was found to be highly conserved by phylogenetic tree analysis, bra-miR167a and ath-miR167a have high homology.
Point 3: Related to this, It is not clear whether the STTM167a lines carry the binding domain of bra-miR167a or ath-miR167a.
Response 3: Thanks for your comments. We have added this content in line 411-415:” The STTM vector was constructed by inserting TAG bases into the 10th and 11th bases to make it silence of the mature miRNA sequence The STTM miR167a interme-diate forms a stem-loop structure on its own after being transcribed into RNA, with both ends present as a single-stranded complementary portion to miR167a, which binds in a complementary manner to miR167a”.
Point 4: The possible effect on ARF6 expression is never mentioned nether tested. Might the overexpression of bra-miR167 in Arabidopsis repress unspecific targets?
Response 4: In previous studies, we mainly found that ARF8 was expressed in different periods after pathogenic infection, and the previous discovery was that miR167-RF8 had an impact on the regulation of the plant's lateral roots. So we focused the main research objects bra-miR167A and ARF8
Point 5: Authors are also missing to report in the introduction about the information already available in the literature regarding the role of ARF8 and ARF6 in adventitious root development. The root phenotype reported in Figure 5 should be implemented with a detailed analysis regarding the effects on the adventitious roots (ref 44 is mentioned only after in the discussion section).
Response 5: Thanks for your comments. We have made modifications this section in line 66-73:” miR167 and its targets ARF6 and ARF8 also regulate the lateral root and adventitious root development. In Arabidopsis, miR167 negatively regulates the formation of adventitious roots in plants, while its target gene ARF6/8 positively regulates the formation of adventitious roots in plants [19]. Meanwhile, ath-miR167 and ARF8 are expressed in the pericycle of root where lateral roots emerge and medicate the development of lateral roots [Kinoshita, 2012 #54]. In soybean, miR167-GmARF8 is regarded as a key regulatory module of root nodule and lateral development [20]”
Point 6: All section 2.2 of the results are mainly material and method information.
Response 6: Thanks for your comments. We have made modifications this section.
Point 7: Line 441,The expression analysis of each gene was per-441 formed with three biological replicates and two technical replicates." Could you explain?
Response 7: Thanks for your comments. I apologize for my mistake. I had already corrected that.
Point 8: Section 2.4 Figure 6. This part should be the most important, but the experiment conducted to test susceptibility to clubroot is poorly explained (how many biological replicates were done on how many plants?) Is the result reported in Fig. 6B statistically significant? Which statistical test was conducted?.
Response 8: Thanks for your comments. We had conducted a more detailed analysis on the disease index and supplemented it this to the materials and methods in line 433-453 “The roots of the plants were washed, and the incidence of disease was critically observed at 30 DAI. STTM167a, OE-miR167a and WT each with three plates and one plate with 50 holes, total 450 plants. Based on the size of the galls, they were graded as follows: grade 0 (no nodule in the lateral and main roots), grade 1 (small nodule in the lateral root), grade 2 (small nodule in the main root), grade 3 (nodule in both the lat-eral and main roots), and grade 4 (nodule in both the lateral and main roots) [66]. The disease index was calculated as follows: DI = (1 n1 + 2 n2 + 3 n3 + 4 n4) × 100/4Nt (n represents the number of diseased plants/grade, N represents the total number of dis-eased plants), DI = 0 (highly resistant), DI < 10 (resistant), 10 ≤ DI < 20 (moderately susceptible), 20 ≤ DI < 50 (susceptible), and DI ≥ 50 (highly susceptible). Each plate calculated a disease index and each line of Arabidopsis thaliana would get three disease indexes.” and in line 463-470:” The miRNA, transcriptome sequencing and phytohormone metabolism data analysis method refer to Xiaochun Wei [37]. All three biological repeated experiments were calculated using IBM SPSS Statistics 2.6. and one-way or two way variance analysis (ANOVA) was performed to determine the significant difference of statistics. The value given is the average ± standard deviation (SD) for three biological replicates and SD value was shown by the error bars in the figures. P value is corrected with Bonferroni, * P <0.05, ** P <0.01, *** P <0.001, **** P <0.0001. Moreover, different letters indicate significant differences at P < 0.05.”

Reviewer 3 Report
The work is generally correct, however, the Results are superficially presented.
What is it DAI? (Figure 6)?
Figure 5: ,,lenght (nm)" ?
The caption are generally not properly described, e.g.
Figure 2: ,,content analysis of susceptible materials at differ-136 ent periods after inoculation." Could you explain it?
However, the some methods are not properly decribed, e.g.
Line 412: ,,proper amount" Could you explain what is it ,,proper".
Line 431: ,,Paraffin sectioning" I don't understand what the Authors mean. This method in not described properly and the publication [61] is insufficient.
Line 441 ,,The expression analysis of each gene was per-441 formed with three biological replicates and two technical replicates." Could you explain?
Statistical methods are not described. Why Tukey's HSD? I doubt is properly in this research. Explain, please.
ETC.
Some sentences are convoluted and difficult to understand.
Author Response
Dear Reviewer,
I would like to extend my gratitude to the respected reviewers, who spend your valuable time to review our manuscript. Your valuable comments were necessarily important for substantial improvement of our manuscript. The given comments and suggestions which we try to address point-by- point as follows:
Point 1: The work is generally correct, however, the Results are superficially presented.
Response 1: Thanks for your comments. We have made modificastions to the results section
Point 2: What is it DAI? (Figure 6)?
Response 2: DAI means days after inoculation, see in line 130.
Point 3: Figure 5: ,,lenght (nm)" ?
Response 3: Thanks for your comments. We apologized for our mistake. We have corrected this error.
Point 4: Figure 2: ,,content analysis of susceptible materials at differ-136 ent periods after inoculation." Could you explain it?
Response 4: Thanks for your comments. We have modified this in line 141-143:” IAA content was measured in treated group which was inoculated with clubroot pathogen and control group was not inoculated for susceptible materials at different periods.”
Point 5: Line 412: ,,proper amount" Could you explain what is it ,,proper".
Response 5: Thanks for your comments. I apologize for my inappropriate expression We have corrected this. See in line 432-434:” The Chinese cabbage swollen root was weighed according to the ratio of target patho-genic fluid volume: swollen root weight = 10:1 from -20 °C refrigerator and poured it into a blender for crushing.”
Point 6: Paraffin sectioning" I don't understand what the Authors mean. This method in not described properly and the publication [61] is insufficient.
Response 6: Thanks for your comments. We have made modifications this section in line 455-466:” Transgenic A. thaliana with STTM167a and OE-miR167a genes and WT plants were observed by paraffin sectioning. The fresh tissue about 1 cm was cut and put it into the fixative solution. Then taken it out form the solution, tissue was flatted and put into the embedding frame. Thereafter placed in dehydrating box into a dehydrator to dehydrator and to wax dipping according a series of treatments (75% 4 h-85% 2 h-90% 2 h-95% 1 h-100% 1 h- alcohol benzene 10 min-xylene 20 min -65 °C 3 h). First, the melted wax was put into the embedding frame and the tissue was removed and placed in the embedding frame. After cooling on a -20 °C freezing platform, the solidified wax was removed from the embedding frame for flatting. The flatted wax block was sliced with a thick-ness of 4 μM. Slices were placed on a spreading machine to flatten the tissue. Slices were placed in a 60 °C oven for baking. The baked wax was removed and stored the sample at room temperature for later use. Slice was put into Toluidine blue dye solution for about 2-5min, then washed with water and finally microscopic inspection was conducted .”
Point 7: Line 441 ,,The expression analysis of each gene was per-441 formed with three biological replicates and two technical replicates." Could you explain?
Response 7: Thanks for your comments. I apologize for my mistake. I had already correctted that.
Point 8: Statistical methods are not described. Why Tukey's HSD? I doubt is properly in this research. Explain, please..
Response 8: Thanks for your comments. I have add the detail method of data analysis in line 495-500:” All three biological repeated experiments are calculated using IBM SPSS Statistics 2.6. and ANOVA was performed to determine the significant difference of statistics. Tukey’s HSD were done for significant differences among the means of different parameters following the principles of multiple mean separation in statistics. The value given is the average ± standard deviation (SD) for three biological replicates and SD value was shown by the error bars in the figures. P value is corrected with Bonferroni, *P <0.05, ** P <0.01, *** P <0.001, **** P <0.0001.”

Round 2
Reviewer 3 Report
The manuscript is corrected sufficiently to be published.